# Container: Context Aggregation Network

**Peng Gao**[1,2], **Jiasen Lu**[4], **Hongsheng Li**[2], **Roozbeh Mottaghi**[3,4], **Aniruddha Kembhavi**[3,4]

[1] Shanghai AI Laboratory     [2] CUHK-SenseTime Joint Lab, CUHK
[3] University of Washington  [4] PRIOR @ Allen Institute for AI

## Abstract

Convolutional neural networks (CNNs) are ubiquitous in computer vision, with a myriad of effective and efficient variations. Recently, Transformers – originally introduced in natural language processing – have been increasingly adopted in computer vision. While early adopters continue to employ CNN backbones, the latest networks are end-to-end CNN-free Transformer solutions. A recent surprising finding shows that a simple MLP based solution without any traditional convolutional or Transformer components can produce effective visual representations. While CNNs, Transformers and MLP-Mixers may be considered as completely disparate architectures, we provide a unified view showing that they are in fact special cases of a more general method to aggregate spatial context in a neural network stack. We present the CONTAINER (CONText AggregatIon NEtwoRk), a general-purpose building block for multi-head context aggregation that can exploit long-range interactions *a la* Transformers while still exploiting the inductive bias of the local convolution operation leading to faster convergence speeds, often seen in CNNs. Our CONTAINER architecture achieves 82.7 % Top-1 accuracy on ImageNet using 22M parameters, +2.8 improvement compared with DeiT-Small, and can converge to 79.9 % Top-1 accuracy in just 200 epochs. In contrast to Transformer-based methods that do not scale well to downstream tasks that rely on larger input image resolutions, our efficient network, named CONTAINER-LIGHT, can be employed in object detection and instance segmentation networks such as DETR, RetinaNet and Mask-RCNN to obtain an impressive detection mAP of 38.9, 43.8, 45.1 and mask mAP of 41.3, providing large improvements of 6.6, 7.3, 6.9 and 6.6 pts respectively, compared to a ResNet-50 backbone with a comparable compute and parameter size. Our method also achieves promising results on self-supervised learning compared to DeiT on the DINO framework. Code is released at https://github.com/allenai/container.

## 1 Introduction

Convolutional neural networks (CNNs) have become the *de facto* standard for extracting visual representations, and have proven remarkably effective at numerous downstream tasks such as object detection [37], instance segmentation [22] and image captioning [1]. Similarly, in natural language processing, Transformers rule the roost [13, 43, 42, 4]. Their effectiveness at capturing short and long range information have led to state-of-the-art results across tasks such as question answering [45] and language understanding [58].

In computer vision, Transformers were initially employed as long range information aggregators across space (e.g., in object detection [5]) and time (e.g., in video understanding [61]), but these methods continued to use CNNs [34] to obtain raw visual representations. More recently however, CNN-free visual backbones employing Transformer modules [54, 14] have shown impressive performance on image classification benchmarks such as ImageNet [33]. The race to dethrone CNNs has now begun to expand beyond Transformers – a recent unexpected result shows that a multi-layer perceptron (MLP) exclusive network [52] can be just as effective at image classification.

35th Conference on Neural Information Processing Systems (NeurIPS 2021).

On the surface, CNNs [34, 8, 63, 23], Vision Transformers (ViTs) [14, 54] and MLP-mixers [52] are typically presented as disparate architectures. However, taking a step back and analyzing these methods reveals that their core designs are quite similar. Many of these methods adopt a cascade of neural network blocks. Each block typically consists of aggregation modules and fusion modules. Aggregation modules share and accumulate information across a predefined context window over the module inputs (e.g., the self attention operation in a Transformer encoder), while fusion modules combine position-wise features and produce module outputs (e.g., feed forward layers in ResNet).

In this paper, we show that the primary differences in many popular architectures result from variations in their aggregation modules. These differences can in fact be characterized as variants of an affinity matrix within the aggregator that is used to determine information propagation between a query vector and its context. For instance, in ViTs [14, 54], this affinity matrix is dynamically generated using key and query computations; but in the Xception architecture [8] (that employs depthwise convolutions), the affinity matrix is static – the affinity weights are the same regardless of position, and they remain the same across all input images regardless of size. And finally the MLP-Mixer [52] also uses a static affinity matrix which changes across the landscape of the input.

Along this unified view, we present CONTAINER (CONText AggregatIon NEtwoRk), a general purpose building block for multi-head context aggregation. A CONTAINER block contains both static affinity as well as dynamic affinity based aggregation, which are combined using learnable mixing coefficients. This enables the CONTAINER block to process long range information while still exploiting the inductive bias of the local convolution operation. CONTAINER blocks are easy to implement, can easily be substituted into many present day neural architectures and lead to highly performant networks whilst also converging faster and being data efficient.

Our proposed CONTAINER architecture obtains 82.7 % Top-1 accuracy on ImageNet using 22M parameters, improving +2.8 points over DeiT-S [54] with a comparable number of parameters. It also converges faster, hitting DeiT-S's accuracy of 79.9 % in just 200 epochs compared to 300.

We also propose a more efficient model, named CONTAINER-LIGHT that employs only static affinity matrices early on but uses the learnable mixture of static and dynamic affinity matrices in the latter stages of computation. In contrast to ViTs that are inefficient at processing large inputs, CONTAINER-LIGHT can scale to downstream tasks such as detection and instance segmentation that require high resolution input images. Using a CONTAINER-LIGHT backbone and 12 epochs of training, RetinaNet [37] is able to achieve 43.8 mAP, while Mask-RCNN [22] is able to achieve 45.1 mAP on box and 41.3 mAP on instance mask prediction, improvements of +7.3, +6.9 and +6.6 respectively, compared to a ResNet-50 backbone. The more recent DETR and its variants SMCA-DETR and Deformable DETR [5, 19, 75] also benefit from CONTAINER-LIGHT and achieve 38.9, 43.0 and 44.2 mAP, improving significantly over their ResNet-50 backbone baselines.

CONTAINER-LIGHT is data efficient. Our experiments show that it can obtain an ImageNet Top-1 accuracy of 61.8 using just 10% of training data, significantly better than the 39.3 accuracy obtained by DeiT. CONTAINER-LIGHT also convergences faster and achieves better kNN accuracy (71.5) compared to DeiT (69.6) under DINO self-supervised training framework [6].

The CONTAINER unification and framework enable us to easily reproduce several past models and even extend them with just a few code and parameter changes. We extend multiple past models and show improved performance – for instance, we produce a Hierarchical DeiT model, a multi-head MLP-Mixer and add a static affinity matrix to the DeiT architecture. Our code base and models will be released publicly. Finally, we analyse a CONTAINER model containing both static and dynamic affinities and show the emergence of convolution-like local affinities in the early layers of the network.

In summary, our contributions include: (1) A unified view of popular architectures for visual inputs – CNN, Transformer and MLP-mixer, (2) A novel network block – CONTAINER, which uses a mix of static and dynamic affinity matrices via learnable parameters and the corresponding architecture with strong results in image classification and (3) An efficient and effective extension – CONTAINER-LIGHT with strong results in detection and segmentation. Importantly, we see that a number of concurrent works are aiming to fuse the CNN and Transformer architectures [36, 64, 40, 24, 55, 69, 64, 47], validating our approach. We hope that our unified view helps place these different concurrent proposals in context and leads to a better understanding of the landscape of these methods.

## 2 Related Work

**Visual Backbones.** Since AlexNet [33] revolutionized computer vision, a host of CNN based architectures have provided further improvements in terms of accuracy including VGG [46], ResNet [23], Inception Net [48], SENet [28], ResNeXt [63] and Xception [8] and efficiency including Mobile-net v1 [26], Mobile-net v2 [26] and Efficient-net v2 [50]. With the success of Transformers [56] in NLP such as BERT [13] and GPT [43], researchers have begun to apply them towards solving the long range information aggregation problem in computer vision. ViT [14]/DeiT [54] are transformers that achieve better performance on ImageNet than CNN counterparts. Recently, several concurrent works explore integrating convolutions with transformers and achieve promising results. ConViT [11] explores soft convolutional inductive bias for enhancing DeiT. CeiT [66] directly incorporates CNNs into the Feedforward module of transformers to enhance the learned features. PVT [60] proposes a pyramid vision transformer for efficient transfer to downstream tasks. Pure Transformer models such as ViT/DeiT however, require huge GPU memory and computation for detection [60] and segmentation [73] tasks, which need high resolution input. MLP-Mixer [52] shows that simply performing transposed MLP followed by MLP can achieve near state-of-the-art performance. We propose CONTAINER, a new visual backbone that provides a unified view of these different architectures and performs well across several vision tasks including ones that require a high resolution input.

**Transformer Variants.** Vanilla Transformers are unable to scale to long sequences or high-resolution images due to the quadratic computation in self-attention. Several methods have been proposed to make Transformer computations more efficient for high resolution input. Reformer [32], Clusterform [57], Adaptive Clustering Transformer [73] and Asymmetric Clustering [10] propose to use Locality Sensitivity Hashing to cluster keys or queries and reduce quadratic computation into linear computation. Lightweight convolution [62] explore convolution architectures for replacing Transformers but only explore applications in NLP. RNN Transformer [31] builds a connection between RNN and Transformer and results in attention with linear computation. Linformer [59] changes the multiplication order of key,query,value into query,value,key by deleting the softmax normalization layer and achieve linear complexity. Performer [9] uses Orthogonal Random Feature to approximate full rank softmax attention. MLIN [18] performs interaction between latent encoded nodes, and its complexity is linear with respect to input length. Bigbird [3] breaks the full rank attention into local, randomly selected and global attention. Thus the computation complexity becomes linear. Longformer [68] uses local Transformers to tackle the problem of massive GPU memory requirements for long sequences. MLP-Mixer [52] is a pure MLP architecture for image recognition. In the unified formulation we provide, MLP-Mixer can be considered as a single-head Transformer with static affinity matrix weight. MLP-Mixer can provide more efficient computation than vanilla transformer due to no need to calculate affinity matrix using key query multiplication. Efficient Transformers mostly use approximate message passing which results in performance deterioration across tasks. Lightweight Convolution [62], Involution [35], Synthesizer [51], and MUSE [71] explored the relationship between Depthwise Convolution and Transformer. Our CONTAINER unification performs global and local information exchange simultaneously using a mixture affinity matrix, while CONTAINER-LIGHT switches off the dynamic affinity matrix for high resolution feature maps to reduce computation. Although switching off the dynamic affinity matrix slightly hinders classification performance, CONTAINER-LIGHT still provides effective and efficient generalization to downstream tasks compared with popular backbones such as ViT and ResNet.

**Transformers for Vision.** Transformers enable high degrees of parallelism and are able to capture long-range dependencies in the input. Thus Transformers have gradually surpassed other architectures such as CNN [34] and RNN [25] on image [14, 5, 70], audio [2], multi-modality [17, 21, 20], and language understanding [13]. In computer vision, Non-local Neural Network [61] has been proposed to capture long range interactions to compensate for the local information captured by CNNs and used for object detection [27] and semantic segmentation [16, 29, 76, 67]. However, these methods use Transformers as a refinement module instead of treating the transformer as a first-class citizen. ViT [14] introduces the first pure Transformer model into computer vision and surpasses CNNs with large scale pretraining on the non publicly available JFT dataset. DeiT [54] trains ViT from scratch on ImageNet-1k and achieve better performance than CNN counterparts. DETR [5] uses Transformer as an encoder and decoder architecture for designing the first end-to-end object detection system. Taming Transformer [15] use Vector Quantization [41] GAN and GPT [43] for high quality high-resolution image generation. Motivated by the success of DETR on object detection, Transformers have been applied widely on tasks such as semantic segmentation [74], pose estimation [65], trajectory

estimation [39], 3D representation learning and self-supervised learning with MOCO v3 [7] and DINO [6]. ProTo [72] verify the effective of transformer on reasoning tasks.

## 3 Methods

In this section we first provide a generalized view of neighborhood/context aggregation modules commonly employed in present neural networks. Then we revisit three major architectures – Transformer [56], Depthwise Convolution [8] and the recently proposed MLP-Mixer [52], and show that they are special cases of our generalized view. We then present our CONTAINER module in Sec 3.3 and its efficient version – CONTAINER-LIGHT in Sec 3.5.

### 3.1 Contextual Aggregation for Vision

Consider an input image $X \in \mathbb{R}^{C \times H \times W}$, where $C$ and $H \times W$ denote the channel and spatial dimensions of the input image, respectively. The input image is first flattened to a sequence of tokens $\{X_i \in \mathbb{R}^C | i = 1, \ldots, N\}$, where $N = HW$ and input to the network. Vision networks typically stack multiple building blocks with residual connections [23], defined as

$$\mathbf{Y} = \mathcal{F}(\mathbf{X}, \{\mathbf{W}_i\}) + \mathbf{X}. \tag{1}$$

Here, $\mathbf{X}$ and $\mathbf{Y}$ are the input and output vectors of the layers considered, and $\mathbf{W}_i$ represents the learnable parameters. $\mathcal{F}$ determines how information across $\mathbf{X}$ is aggregated to compute the feature at a specific location. We first define an affinity matrix $\mathcal{A} \in \mathbb{R}^{N \times N}$ that represents the neighborhood for contextual aggregation. Equation 1 can be re-written as:

$$\mathbf{Y} = (\mathcal{A}\mathbf{V})\mathbf{W}_1 + \mathbf{X}, \tag{2}$$

where $\mathbf{V} \in \mathbb{R}^{N \times C}$ is a transformation of $\mathbf{X}$ obtained by a linear projection $\mathbf{V} = \mathbf{X}\mathbf{W}_2$. $\mathbf{W}_1$ and $\mathbf{W}_2$ are the learnable parameters. $\mathcal{A}_{ij}$ is the affinity value between $X_i$ and $X_j$. Multiplying the affinity matrix with $\mathbf{V}$ propagates information across features in accordance with the affinity values.

The modeling capacity of such a context aggregation module can be increased by introducing multiple affinity matrices, allowing the network to have several pathways to contextual information across $\mathbf{X}$. Let $\{\mathbf{V}^i \in \mathbb{R}^{N \times \frac{C}{M}} | i = 1, \ldots, M\}$ be slices of $\mathbf{V}$, where $M$ is the number of affinity matrices, also referred to as the number of heads. The multi-head version of Equation 2 is

$$\mathbf{Y} = \text{Concat}(\mathcal{A}_1\mathbf{V}_1, \ldots, \mathcal{A}_M\mathbf{V}_M)\mathbf{W}_2 + \mathbf{X}, \tag{3}$$

where $\mathcal{A}_m$ denotes the affinity matrix in each head. Different $A_m$ can potentially capture different relationships within the feature space and thus increase the representation power of contextual aggregation compared with a single-head version. Note that only spatial information is propagated during contextual aggregation using the affinity matrices; cross-channel information exchange does not occur within the affinity matrix multiplication, and that there is no non-linear activation function.

### 3.2 The Transformer, Depthwise Convolution and MLP-Mixer

Transformer [56], depthwise convolution [30] and the recently proposed MLP-Mixer [52] are three distinct building blocks used in computer vision. Here, we show that they can be represented within the above context aggregation framework, by defining different types of affinity matrices.

**Transformer.** In the self-attention mechanism in Transformers, the affinity matrix is modelled by the similarity between the projected query-key pairs. With $M$ heads, the affinity matrix in head $m$, $\mathcal{A}_m^{sa}$ can be written as

$$\mathcal{A}_m^{sa} = \text{Softmax}(\mathbf{Q}_m\mathbf{K}_m^T / \sqrt{C/M}), \tag{4}$$

where $\mathbf{K}_m, \mathbf{Q}_m$ are the corresponding key, query in head $m$, respectively. The affinity matrix in self-attention is dynamically generated and can capture instance level information. However, this introduces quadratic computational, which requires heavy computation for high resolution feature.

**Depthwise Convolution.** The convolution operator fuses both spatial and channel information in parallel. This is different from the contextual aggregation block defined above. However, depthwise convolution [30] which is an extreme case of group convolution performs disentangled convolution.

Considering the number of the heads from the contextual aggregation block to be equal to the channel size $C$, we can define the convolutional affinity matrix given the 1-d kernel $Ker \in \mathbb{R}^{C \times 1 \times k}$:

$$\mathcal{A}^{conv}_{mij} = \begin{cases} Ker[m, 0, |i-j|] & |i-j| \leq k \\ 0 & |i-j| > k \end{cases}, \tag{5}$$

where $\mathcal{A}_{mij}$ is the affinity value between $X_i$ and $X_j$ on head $m$. In contrast with the affinity matrix obtained from self-attention whose value is conditioned on the input feature, the affinity values for convolution are static – they do not depend on the input features, sparse – only involves local connections and shared across the affinity matrix.

**MLP-Mixer**   The recently proposed MLP-Mixer [52] does not rely on any convolution or self-attention operator. The core of MLP-Mixer is the transposed MLP operation, which can be denoted as $\mathbf{X} = \mathbf{X} + (\mathbf{V}^T \mathbf{W}_{MLP})^T$. We can define the affinity matrix as

$$\mathcal{A}^{mlp} = (\mathbf{W}_{MLP})^T, \tag{6}$$

where $\mathbf{W}_{MLP}$ represents the learnable parameters. This simple equation shows that the transposed-MLP operator is a contextual aggregation operator on a single feature group with a dense affinity matrix. Comparing with self-attention and depthwise convolution, the transpose-MLP affinity matrix is static, dense and with no parameter sharing.

The above simple unification reveals the similarities and differences between Transformer, depthwise convolution and MLP-Mixer. Each of these building blocks can be obtained by different formulating different affinity matrices. This finding leads us to create a powerful and efficient building block for vision tasks – the CONTAINER.

### 3.3   The CONTAINER Block

As detailed in Sec 3.2, previous architectures have employed either static or dynamically generated affinity matrices – each of which provides its unique set of advantages and features. Our proposed building block named CONTAINER, combines both types of affinity matrices via a learnable parameter. The single head CONTAINER is defined as:

$$\mathbf{Y} = ((\alpha \overbrace{\mathcal{A}(\mathbf{X})}^{Dynamic} + \beta \overbrace{\mathcal{A}}^{Static})V)W_2 + \mathbf{X} \tag{7}$$

$\mathcal{A}(\mathbf{X})$ is dynamically generated from $\mathbf{X}$ while $\mathcal{A}$ is a static affinity matrix. We now present a few special cases of the CONTAINER block. In the following, $\mathcal{L}$ denotes a learnable parameter.

- $\alpha = 1$, $\beta = 0$, $\mathcal{A}(x) = \mathcal{A}^{sa}$: A vanilla Transformer block with self-attention (denoted $sa$).

- $\alpha = 0$, $\beta = 1$, $M = C$, $\mathcal{A} = \mathcal{A}^{conv}$: A depthwise convolution block. In depthwise convolution, each channel has a different static affinity matrix. When $M \neq C$, the resultant block can be considered a Multi-head Depthwise Convolution block (MH-DW). MH-DW shares kernel weights.

- $\alpha = 0$, $\beta = 1$, $M = 1$, $\mathcal{A} = \mathcal{A}^{mlp}$: An MLP-Mixer block. When $M \neq 1$, we name the module Multi-head MLP (MH-MLP). MH-MLP splits channels into $M$ groups and performs independent transposed MLP to capture diverse static token relationships.

- $\alpha = \mathcal{L}$, $\beta = \mathcal{L}$, $\mathcal{A}(x) = \mathcal{A}^{sa}$, $\mathcal{A} = \mathcal{A}^{mlp}$: This CONTAINER block fuses dynamic and static information, but the static affinity resembles the MLP-Mixer matrix. We call this block CONTAINER-PAM (Pay Attention to MLP).

- $\alpha = \mathcal{L}$, $\beta = \mathcal{L}$, $\mathcal{A}(x) = \mathcal{A}^{sa}$, $\mathcal{A} = \mathcal{A}^{conv}$: This CONTAINER block fuses dynamic and static information, but the static affinity resembles the depthwise convolution matrix. This static affinity matrix contains a locality constraint which is shift invariant, making it more suitable for vision tasks. This is the default configuration used in our experiments.

The CONTAINER block is easy to implement and can be readily swapped into an existing neural network. The above versions of CONTAINER provide variations on the resulting architecture and its performance and exhibit different advantages and limitations. The computation cost of a CONTAINER block is the same as a vanilla Transformer since the static and dynamic matrices are linearly combined.

## 3.4 The CONTAINER network architecture

We now present a base architecture used in our experiments. The unification of past works explained above allows us to easily compare self-attention, depthwise convolution, MLP and multiple variations of the CONTAINER block, and we perform these comparison using a consistent base architecture.

Motivated by networks in past works [23, 60], our base architecture contains 4 stages. In contrast to ViT/DeiT which down-sample the image to a low resolution and keep this resolution constant, each stage in our architecture down-samples the image resolution gradually. Gradually down-sampling can retain image details, which is important for downstream tasks such as segmentation and detection. Each of the 4 stages contains a cascade of blocks. Each block contains two sub-modules, the first to aggregate spatial information (named spatial aggregation module) and the second to fuse channel information (named feed-forward module). In this paper, the channel fusion module is fixed to a 2-layer MLP as proposed in [56]. Designing a better spatial aggregation module is the main focus of this paper. The 4 stages contain 2, 3, 8 and 3 blocks respectively. Each stage uses patch embeddings which fuse spatial patches of size $p \times p$ into a single vector. For the 4 stages, the values of $p$ are 4,4,2,2 respectively. The feature dimension within a stage remains constant – and is set to 128, 256, 320, and 512 for the four stages. This base architecture augmented with the CONTAINER block results in a similar parameter size as DeiT-S [54].

## 3.5 The CONTAINER-LIGHT network

We also present an efficient version known as CONTAINER-LIGHT which uses the same basic architecture as CONTAINER, but switches off the dynamic affinity matrix in the first 3 stages. The absence of the computation heavy dynamic attention at the early stages of computation help efficiently scale the model to process large image resolutions and achieve superior performance on downstream tasks such as detection and instance segmentation.

$$\mathcal{A}_m^{\text{CONTAINER-LIGHT}} = \begin{cases} \mathcal{A}_m^{conv} & Stage = 1, 2, 3 \\ \alpha \mathcal{A}_m^{sa} + \beta \mathcal{A}_m^{conv} & Stage = 4 \end{cases}, \tag{8}$$

$\alpha$ and $\beta$ are learnable parameters. In network stage 1, 2, 3, CONTAINER-LIGHT will switch off $\mathcal{A}_m^{sa}$.

# 4 Experiments

We now present experiments with CONTAINER for ImageNet and with CONTAINER-LIGHT for the tasks of object detection, instance segmentation and self-supervised learning. We also present appropriate baselines. Please see the appendix for details of the models, training and setup.

## 4.1 ImageNet Classification

**Top-1 Accuracy.** Table 1 compares several highly performant models within the CNN, Transformer, MLP, Hybrid and our proposed CONTAINER families. CONTAINER and CONTAINER-LIGHT outperform the pure Transformer models ViT [14] and DeiT [54] despite far fewer parameters. They outperform PVT [60] which employ a hierarchical representation similar to our base architecture. They also outperform the recently published state-of-the-art SWIN [40] (they outperform Swin-T which has more parameters). The best performing models continue to be from the EfficientNet [49] family, but we note that EfficientNet [49] and RegNet [44] apply an extensive neural architecture search, which we do not. Finally note that CONTAINER-LIGHT not only achieves a high accuracy but does so at lower FLOPs and much faster throughput than models with comparable capacities.

The CONTAINER framework allows us to easily reproduce past architectures but also to create effective extensions over past work (outlined in Sec 3.3), several of which are compared in Table 2. H-DeiT-S is a hierarchical version of DeiT-S obtained by simply using $\mathcal{A}^{sa}$ within our hierarchical architecture and provides 1.2 gain. Conv-3 (naive convolution (conv) with $3 \times 3$ kernel) aggregates spatial and channel information, where as Group Conv-3 splits input features and performs convs using different kernels – it is cheaper and more effective. When group size = channel dim., we get depth-wise conv. DW-3 is a depthwise convs with 3 by 3 kernel that only aggregates spatial information. Channel information is fused using $1 \times 1$ convs. MH-DW-3 is a multi-head version of DW-3. MH-DW-3 shares kernel parameters within the same group. With fewer kernels, MH-DW-3 achieves comparable performance with DW-3. MLP is an implementation of transposed MLP for spatial propagation. MLP-LR stands for MLP with low-rank decompostion. MLP-LR provides better

| Family | Network | Top-1 Acc | Params | FLOPs | Throughput | Input dim | NAS |
|---|---|---|---|---|---|---|---|
| | ResNet-50 [23] | 78.5 | 25.6M | 4.1G | 1250.3 | $224^2$ | ✗ |
| | ResNet-101 [23] | 79.8 | 44.7M | 7.9G | 753.7 | $224^2$ | ✗ |
| | Xception71 [8] | 79.9 | 42.3M | N/A | 423.5 | $299^2$ | ✗ |
| | RegNetY-4G [44] | 80.0 | 21M | 4.0G | 1156.7 | $224^2$ | ✓ |
| | RegNetY-8G [44] | 81.7 | 39M | 8.0G | 591.6 | $224^2$ | ✓ |
| CNN | RegNetY-16G [44] | 82.9 | 84M | 16.0G | 334.7 | $224^2$ | ✓ |
| | EfficientNet-B3 [49] | 81.6 | 12M | 1.8G | 732.1 | $300^2$ | ✓ |
| | EfficientNet-B4 [49] | 82.9 | 19M | 4.2G | 349.4 | $380^2$ | ✓ |
| | EfficientNet-B5 [49] | 83.6 | 30M | 9.9G | 169.1 | $456^2$ | ✓ |
| | EfficientNet-B6 [49] | 84.0 | 43M | 19.0G | 96.9 | $528^2$ | ✓ |
| | EfficientNet-B7 [49] | 84.3 | 66M | 37.0G | 55.1 | $600^2$ | ✓ |
| | ViT-B/16 [14] | 77.9 | 86M | 55.4G | 85.9 | $384^2$ | ✗ |
| | ViT-L/16 [14] | 76.5 | 307M | 190.7G | 27.3 | $384^2$ | ✗ |
| | DeiT-S [54] | 79.9 | 22.1M | 4.6G | 940.4 | $224^2$ | ✗ |
| | DeiT-B [54] | 81.8 | 86M | 17.5G | 292.3 | $224^2$ | ✗ |
| | PVT-T [60] | 75.1 | 13.2M | 1.9G | N/A | $224^2$ | ✗ |
| | PVT-S [60] | 79.8 | 24.5M | 3.8G | N/A | $224^2$ | ✗ |
| Transformer | PVT-Medium [60] | 81.2 | 44.2M | 6.7G | N/A | $224^2$ | ✗ |
| | PVT-L [60] | 81.7 | 61.4M | 9.8G | N/A | $224^2$ | ✗ |
| | ViL-T [69] | 76.3 | 6.7M | 1.3G | N/A | $224^2$ | ✗ |
| | ViL-S [69] | 82.0 | 24.6M | 4.9G | N/A | $224^2$ | ✗ |
| | Swin-T [40] | 81.3 | 29M | 4.5G | 755.2 | $224^2$ | ✗ |
| | Swin-S [40] | 83.0 | 50M | 8.7G | 436.9 | $224^2$ | ✗ |
| | Swin-B [40] | 83.3 | 88M | 15.4G | 278.1 | $224^2$ | ✗ |
| MLP | Mixer-B/16 [52] | 76.4 | 79M | N/A | N/A | $224^2$ | ✗ |
| | ResMLP-24 [53] | 79.4 | 30M | 6.0G | 715.4 | $224^2$ | ✗ |
| Hybrid | ConvViT [11] | 81.3 | 27M | 5.4G | N/A | $224^2$ | ✗ |
| | BoT-S1-50 [47] | 79.1 | 20.8M | 4.3G | N/A | $224^2$ | ✗ |
| | BoT-S1-59 [47] | 81.7 | 33.5M | 7.3G | N/A | $224^2$ | ✗ |
| Container | CONTAINER | 82.7 | 22.1M | 8.1G | 347.8 | $224^2$ | ✗ |
| (Ours) | CONTAINER-LIGHT | 82.0 | 20.0M | 3.2G | 1156.9 | $224^2$ | ✗ |

Table 1: ImageNet [12] Top-1 accuracy comparison for CNN, Transformer, MLP, Hybrid and Container models. Throughput (images/s) is not reported in all papers (noted as N/A). Models that have fewer parameters than CONTAINER or upto 10% more parameters are highlighted.

performance with fewer parameters. MH-MLP-LR adds a multi-head mechanism over MLP-LR and provides further improvements. In contrast to the original MLP-Mixer [52], we do not add any non-linearity like GELU into CONTAINER as is specified in the contextual aggregation equation.

**Data Efficiency.** CONTAINER-LIGHT has a built-in shift-invariance and parameter sharing mechanism. As a result it is more data efficient in comparison to DeiT [54]. Table 3 shows that at the low data regime of 10%, CONTAINER-LIGHT outperforms DeiT by a massive 22.5 points.

**Convergence Speed.** Figure 1 (left) compares the convergence speeds of the two CONTAINER variants with a CNN and Transformer (DeiT) [54]. The inductive biases in the CNN enable it to converge faster than DeiT [54], but they eventually perform similarly at 300 epochs, suggesting that dynamic, long range context aggregation is powerful but slow to converge. CONTAINER combines the best of both and provides accuracy improvements with fast convergence. CONTAINER-LIGHT converges as fast with a slight accuracy drop.

| Data ratio | CONTAINER-LIGHT | DeiT |
|---|---|---|
| 100 % | 82.0 (+2.1) | 79.9 |
| 80 % | 81.1 (+2.6) | 78.5 |
| 50 % | 78.8 (+4.8) | 74.0 |
| 10 % | 61.8 (+22.5) | 39.3 |

Table 3: ImageNet Top-1 Acc for CONTAINER-LIGHT and DeiT-S with varying training sizes.

**Emergence of locality.** Within our CONTAINER framework, we can easily add a static affinity matrix to the DeiT architecture. This simple change (1 line of code addition), can provide a +0.5 Top-1

| Method | Top-1 Acc | Params | $\alpha$ | $\beta$ | $\frac{C}{M}$ | $\mathcal{A}^{dynamic}$ | $\mathcal{A}^{static}$ |
|---|---|---|---|---|---|---|---|
| H-DeiT-S | 81.0 | 22.1M | 1 | 0 | 32 | $\mathcal{A}^{sa}$ | N/A |
| Conv-3 | 79.6 | 33.8M | N/A | N/A | N/A | N/A | N/A |
| Group Conv-3 | 79.7 | 20.5M | N/A | N/A | N/A | N/A | N/A |
| DW-3 | 80.1 | 18.7M | 0 | 1 | 1 | N/A | $\mathcal{A}^{conv}$ |
| MH-DW-3 | 79.9 | 18.6M | 0 | 1 | 32 | N/A | $\mathcal{A}^{conv}$ |
| MLP | 77.5 | 50.9M | 0 | 1 | C | N/A | $\mathcal{A}^{mlp}$ |
| MLP-LR | 78.9 | 36.5M | 0 | 1 | C | N/A | $\mathcal{A}^{mlp}$ |
| MH-MLP-LR | 79.6 | 41.6 M | 0 | 1 | 32 | N/A | $\mathcal{A}^{mlp}$ |
| CONTAINER | 82.7 | 22.1M | $\mathcal{L}$ | $\mathcal{L}$ | 32 | $\mathcal{A}^{sa}$ | $\mathcal{A}^{conv}$ |
| CONTAINER-LIGHT | 82.0 | 20.0M | $\mathcal{L}$ | $\mathcal{L}$ | 32 | $\mathcal{A}^{sa}$ | $\mathcal{A}^{conv}$ |

Table 2: ImageNet accuracies for architecture variations (with convolutions, self-attention and MLP) enabled within the CONTAINER framework. As per our notation, $C$: num channels, $M$: num heads, $C/M$: head dimension. See Sec 3.3 and 4.1 for notation and model details.

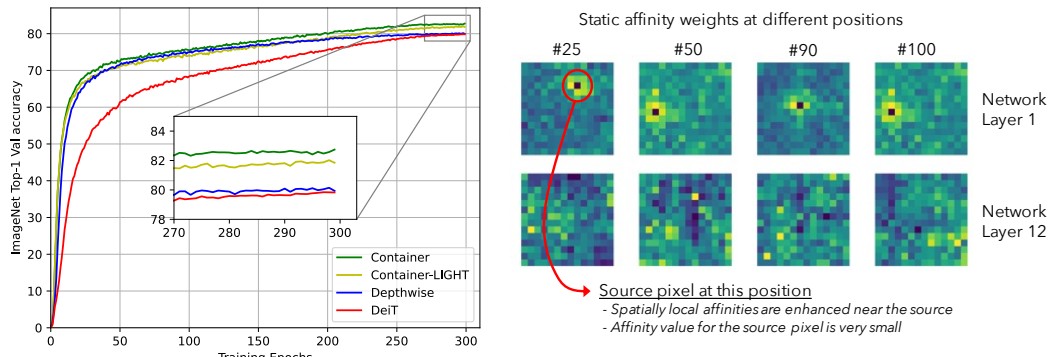

Figure 1: **(left)** Convergence speed comparison between CONTAINER, CONTAINER-LIGHT, Depthwise conv and DeiT. **(right)** Visualization of the static affinity weights at different positions and layers. Layer 1 displays the emergence of local affinities (resembling convolutions).

improvement from 79.9% to 80.4%. This suggests that static and dynamic affinity matrices provide complementary information. As noted in Sec 3.3, we name this CONTAINER-PAM.

It is interesting to visualize the learnt static affinities at different network layers. Figure 1 (right) displays these for 2 layers. Each matrix represents the static affinities for a single position, reshaped to a 2-d grid to resemble the landscape of the neighboring regions. Within Layer 1, we interestingly observe the emergence of local operations via the enhancement of affinity values next to the source pixel (location). These are akin to convolution operations. Furthermore, the affinity value for the source pixel is very small, i.e. at each location, the context aggregator does not use its current feature. We hypothesize that this is a result of the residual connection [23], thereby alleviating the need to include the source feature within the context. Note that in contrast to dynamic affinity, the learnt static matrix is shared for all input images. Notice that Layer 12 displays a more global affinity matrix without any specific interpretable local patterns.

## 4.2 Detection with RetinaNet

Since the attention complexity for CONTAINER-LIGHT is linear at high image resolutions (initial layers) and then quadratic, it can be employed for downstream tasks such as object detection which usually require high resolution feature maps. Table 4 compares several backbones applied to the RetinaNet detector [37] on the COCO dataset [38]. Compared to the popular ResNet-50 [23], CONTAINER-LIGHT achieves 43.8 mAP, an improvement of 7.0, 7.2 and 10.4 on $AP_S$, $AP_M$, and $AP_L$ with comparable parameters and cost. The significant increase for large objects shows the benefits of global attention via the dynamic global affinity matrix in our model. CONTAINER-LIGHT also surpasses the large convolution-based backbone X-101-64 [63] and pure Transformer models with similar number of parameters such as PVT-S [60], ViL-S [69], and SWIN-T [40] by large

| Method | | | Mask R-CNN | | | | | | | | RetinaNet | | | |
|---|---|---|---|---|---|---|---|---|---|---|---|---|---|---|
| | #P | FLOPs | $AP^b$ | $AP^b_{50}$ | $AP^b_{75}$ | $AP^m$ | $AP^m_{50}$ | $AP^m_{75}$ | #P | FLOPs | mAP | $AP_S$ | $AP_M$ | $AP_L$ |
| ResNet50 [23] | 44.2 | 180G | 38.2 | 58.8 | 41.4 | 34.7 | 55.7 | 37.2 | 37.7 | 239G | 36.5 | 20.4 | 40.3 | 48.1 |
| ResNet101 [23] | 63.2 | 259G | 40.0 | 60.5 | 44.0 | 36.1 | 57.5 | 38.6 | 56.7 | 319G | 38.5 | 21.7 | 42.8 | 50.4 |
| X-101-32 [63] | 62.8 | 259G | 41.9 | 62.5 | 45.9 | 37.5 | 59.4 | 40.2 | 56.4 | 319G | 39.9 | 22.3 | 44.2 | 52.5 |
| X-101-64 [63] | 101.9 | 424G | 42.8 | 63.8 | 47.3 | 38.4 | 60.6 | 41.3 | 95.5 | 483G | 41.0 | 23.9 | 45.2 | 54.0 |
| PVT-S [60] | 44.1 | 245G | 40.4 | 62.9 | 43.8 | 37.8 | 60.1 | 40.3 | 34.2 | 226G | 40.4 | 25.0 | 42.9 | 55.7 |
| ViL-S [60] | 45.0 | 174G | 41.8 | 64.1 | 45.1 | 38.5 | 61.1 | 41.4 | 35.6 | 252G | 41.6 | 24.9 | 44.6 | 56.2 |
| SWIN-T [40] | 48.0 | 267G | 43.7 | 66.6 | 47.4 | 39.8 | 63.6 | 42.7 | 385 | 244G | 41.5 | 26.4 | 45.1 | 55.7 |
| ViL-M [60] | 60.1 | 261G | 43.4 | 65.9 | 47.0 | 39.7 | 62.8 | 42.1 | 50.7 | 338G | 42.9 | 27.0 | 46.1 | 57.2 |
| ViL-B [60] | 76.1 | 365G | 45.1 | 67.2 | 49.3 | 41.0 | 64.3 | 44.2 | 66.7 | 443G | 44.3 | 28.9 | 47.9 | 58.3 |
| BoT50 [47] | 39.5 | N/A | 39.4 | 60.3 | 43.0 | 35.3 | 57 | 37.5 | N/A | N/A | N/A | N/A | N/A | N/A |
| BoT50-(6x) [47] | 39.5 | N/A | 43.7 | 64.7 | 47.9 | 38.7 | 61.8 | 41.1 | N/A | N/A | N/A | N/A | N/A | N/A |
| CONTAINER-LIGHT | 39.6 | 237G | 45.1 | 67.3 | 49.5 | 41.3 | 64.2 | 44.5 | 29.7 | 218G | 43.8 | 27.4 | 47.5 | 58.5 |

Table 4: Comparing the CONTAINER-LIGHT backbone with several previous methods at the tasks of object detection and instance segmentation using the Mask-RCNN and RetinaNet networks.

margins. Compared to large Transformer backbones such as ViL-M [69] and ViL-B [69], we achieve comparable performance with significantly fewer parameters and FLOPs.

### 4.3 Detection and Segmentation with Mask-RCNN

Table 4 also compares several backbones for detection and instance segmentation using the Mask R-CNN network [22]. As with the findings for RetinaNet [37], CONTAINER-LIGHT outperforms convolution and Transformer based approaches such as ResNet [23], X-101 [63], PVT [60], ViL [69] and recent state-of-the-art SWIN-T [40] and the recent hybrid approach BoT [47]. It obtains comparable numbers to the much larger ViL-B [69].

### 4.4 Detection with DETR

Table 5 shows that our model can consistently improve object detection performance compared to a ResNet-50 [23] backbone (comparable parameters and computation) on end-to-end object detection using DETR [5]. We demonstrate large improvements with DETR [5], DDETR [75] as well as SMCA-DETR [19]. See appendix for $AP^S$, $AP^M$, and $AP^L$ numbers. All models in table 5 are trained using a 50 epochs schedule.

| Method | mAP |
|---|---|
| DETR-ResNet50 [5] | 32.3 |
| DETR-CONTAINER-LIGHT | 38.9 |
| DDETR w/o multi-scale-ResNet50 [75] | 39.3 |
| DDETR w/o multi-scale-CONTAINER-LIGHT | 43.0 |
| SMCA w/o multi-scale-ResNet50 [19] | 41.0 |
| SMCA w/o multi-scale-CONTAINER-LIGHT | 44.2 |

Table 5: CONTAINER-LIGHT and ResNet-50 backbones with DETR and variants for object detection.

### 4.5 Self supervised learning

We train DeiT [54] and CONTAINER-LIGHT for 100 epochs at the self supervised task of visual representation learning using the DINO framework [6]. Table 6 compares top-10 kNN accuracy for both backbones at different epochs of training. CONTAINER-LIGHT significantly outperforms DeiT with large improvements initially demonstrating more efficient learning.

| Epochs $\rightarrow$ | 20 | 40 | 60 | 80 | 100 |
|---|---|---|---|---|---|
| DeiT [6] | 52.0 | 63.3 | 66.5 | 68.9 | 69.6 |
| CONTAINER-LIGHT | 58.0 | 67.0 | 70.0 | 71.1 | 71.5 |

Table 6: CONTAINER-LIGHT and DeiT on DINO self-supervised learning.

## 5 Conclusion

In this paper, we have shown that disparate architectures such as Transformers, depth-wise CNNs and MLP-based methods are closely related via an affinity matrix used for context aggregation. Using this view, we have proposed CONTAINER, a generalized context aggregation building block that combines static and dynamic affinity matrices using learnable parameters. Our proposed networks, CONTAINER and CONTAINER-LIGHT show superior performance at image classification, object detection, instance segmentation and self-supervised representation learning. We hope that this unified view can motivate future research in the design of effective and efficient visual backbones.

**Limitations**: CONTAINER is very effective at image classification but cannot be directly applied to high resolution inputs. The efficient version CONTAINER-LIGHT, can be used for a variety of tasks. However, its limitation is that it is partially hand-crafted – the dynamic affinity matrix is switched off in the first 3 stages. Future work will address how to learn this using the task at hand.

**Negative societal impact**: This research does not have a direct negative societal impact. However, we should be aware that powerful neural networks, particularly image classification networks can be used for harmful applications like face and gender recognition.

**Disclosure of Funding** This work was partially supported by the Shanghai Committee of Science and Technology, China (Grant No. 21DZ1100100 and 20DZ1100800).

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
