# OpenReview forum: "Container: Context Aggregation Networks"
_NeurIPS.cc/2021/Conference — NeurIPS 2021 Poster_

### Official Review · Reviewer_sZdD · 2021-07-10

**Rating:** 7
**Confidence:** 5

**Summary:**

This paper presents a unified view of CNNs, Transformers and MLP-mixer, and proposes the CONTAINER (CONText AggregatIon NEtwoRk), a general-purpose building block for multi-head context aggregation that combines convolution, attention, and depth-wise MLP. Various experiments on ImageNet classification, object detection with RetinaNet, Mask R-CNN, and DETR, as well as self-supervised learning.

**Limitations And Societal Impact:**

Overall this paper is acceptable. The reviewer expects to see more discussion about the relation, like "Demystifying Local Vision Transformer: Sparse Connectivity, Weight Sharing, and Dynamic Weight" https://arxiv.org/abs/2106.04263. This will help researchers deeply understand transformer attention.

**Main Review:**

Originality: The connection between depth-wise convolution, attention, and channel-wise MLP is interesting. I believe it is novel when this paper is submitted to NeurIPS 2021, and there is a concurrent work: "Demystifying Local Vision Transformer: Sparse Connectivity, Weight Sharing, and Dynamic Weight" https://arxiv.org/abs/2106.04263, providing more studies about relation.

The study will motivate researchers to deeply think about what transformer attention really brings. This paper provides some insights about the relation between transformer attention, MLP and depth-wise convolution. This is a good start though the combination is a little straightforward. Thus I was positive to this paper.

Clarity and quality: This paper is clearly written. The reviewer can easily understand this paper. The results are also good for supporting the claim.

Significance: The reviewer believes that the relation between depth-wise convolution, attention, and channel-wise MLP is more significant than the combination.

**Time Spent Reviewing:**

1 hour

---

> ### Author Response · Authors · 2021-08-10
> **Response to Reviewer sZdD**
>
> Thanks so much for pointing out the concurrent work Demystifying[1]. Demystifying is a highly related concurrent work. The similarity between Demysting and Container are listed below:
>
> Similarities: Both papers are trying to build the relationship between previous operators. Local-global, static-dynamic, weight sharing mechanisms are the core components for connecting popular operators like local attention, attention, dynamic convolution, transformer, and mlp-mixer. Besides, both Demystifying and Container focus on architecture with separate spatial and channel information aggregation.
>
> Differences: Demystifying paper emphasis on window selection which has not been discussed in Container. The Demystifying paper focuses on the performance similarity between depthwise, dynamic convolution, and local transformer. However, a new mechanism called mixture of dynamic and static has been motivated by this observation and demonstrates a much better performance design than SWIN, PVT, and other transformer variants in our Container.
>
> The relationship between Container and Demystifying will be discussed in detail in the updated version.
>
> [1] Demystifying Local Vision Transformer: Sparse Connectivity, Weight Sharing, and Dynamic Weight, Han, Qi, arxiv 2021

---

### Official Review · Reviewer_wcp4 · 2021-07-14

**Rating:** 7
**Confidence:** 4

**Summary:**

The paper proposes a unified view between CNNs (but really just depthwise CNNs), ViTs, and MLP-Mixer models. In essence this unified view consists of having a*static + b*dynamic, where "static" is an input-independent learnable filter (for CNN and MLP-Mixer) and "dynamic" is an input-dependent filter as in self-attention, a/b are "switches".

In addition to this container model, the paper also suggests a "light" variant, where b=0 for all but the last stage, "stage" being the resnet-like definition. This variant is especially good for high-resolution tasks.

**Limitations And Societal Impact:**

Yes

**Main Review:**

I think this is an overall solid paper. The unifying view is not ground-breaking or even surprising, but it's good to investigate it, and the resulting model is thoroughly benchmarked across multiple tasks. I really like the related work, it seems pretty thorough to me.

The model performs well, but agian, it's not ground-breaking: compared to models with similar throughput in Table1, container reaches very similar accuracy.

Two things got me wondering:
1) Container-light seems *very* similar to ResNet50. The throughput is also very similar. However, performance is much better. My guess is this is due to the training recipe and not the model. Hence, it would be good to also include a more modern ResNet for comparison, such as ResNet-RS or NFNet.

2) I must be missing something, but the model consists of 4 stages with downsampling, similar to ResNet. However, the filters shown in Figure1/right have the same size for both the first and the 12th layer? How can this be, shouldn't the one from the 12th layer be lower resolution?

Besides that, I have a few nitpicks:

3) related work: DETR is mentioned as if it were a pure transformer, but it actually uses a ResNet encoder.
4) Table3 is not very useful. There are many more papers on 10% ImageNet with much better results than DeiT. Even ignoring unlabeled data (as used by SimCLR, MoCo, ...), the S4L paper shows a ResNet50 trained on 10% gets 56%. The 50% and 80% ImageNet are almost pointless settings, but 1% is also widespread and missing.
5) l54: what does "the landscape of the input" mean?
6) l303: It's a stretch to call them akin to convolutions, since they don't actually apply across locations. The point is that they are mostly local.

**Time Spent Reviewing:**

3

---

> ### Author Response · Authors · 2021-08-10
> **Response to Reviewer wcp4**
>
> Q1: Container-light seems very similar to ResNet50. The throughput is also very similar. However, performance is much better. My guess is this is due to the training recipe and not the model. Hence, it would be good to also include a more modern ResNet for comparison, such as ResNet-RS or NFNet.
>
> A1: The success of Container is attributed to the deep understanding of the relationship between convolution, transformer, and MLP-Mixer. The container can achieve much better performance than ResNet 50 due to the following reasons.
>
> 1) Decoupling Spatial and Channel fusion. In the Container module, spatial fusion and channel fusion are performed separately. This strategy saves parameters, makes training the network easy, and reduces overfitting. In ResNet50, the vanilla convolution simultaneously learns channel and spatial mixing which makes training hard and prone to overfitting.
>
> 2) Container uses hybrid static and dynamic affinity matrices. Both database-level and instance-level information can be captured. While in ResNet 50, the channel and spatial aggregation are static during inference and shared for all instances.
>
> 3) Another reason is due to the different training recipes mentioned by the reviewer. Our Container follows the same training strategy as single-scale Deit and multi-scale PVT/Swin. The improved parameters/FLOPs/throughput/Accuracy suggests the superiority of the Container module. Besides, we compare with NFNet mentioned by the reviewer. The comparison is listed below. We will update the comparison in the revision.
>
>
> |   Model   |  NF-F0[1]|  NF-F1[1]   | NF-F2[1] |  NF-F3[1] | NF-F4[1] | NF-F5[1] | Container-Light | Container-M|
> | :-: | :-: | :-: | :-:| :-: | :-: | :-: | :-:|:-:|
> |FLOPs(B) |   12.38 |  35.54 | 62.59 | 114.76 | 215.24 | 289.76  |  3.2    |     11.4|
> |Param(M)  |   71.5 |  132.6 | 193.8  | 254.9  | 316.1 |  377.2   |   20       |    57|
> |Throughput| 494.9 |  176.3 | 100.1 |   54.3  |   30.1  |   20.8   |  1156.9  |   340.5|
> |Accuracy  |   83.6  |   84.7  |  85.1   |  85.7  |   85.9  |   86.0    |   82        |   84.5|
>
> Note that Container-M is a large version of Container-S required by the reviewer ecxB in Q8.
>
> Q2: the model consists of 4 stages with downsampling, similar to ResNet. However, the filters shown in Figure1/right have the same size for both the first and the 12th layer? How can this be, shouldn't the one from the 12th layer be lower resolution?
>
> A2:  In Line 296, we mention that Container-PAM uses the DeiT architecture which means the resolution of attention is the same for all layers. Other than this part, all models use a hierarchical representation. We will emphasize this part in the updated version.
>
>
> Q3: DETR is mentioned as if it were a pure transformer, but it actually uses a ResNet encoder.
>
> A3: We will mention DETR uses a ResNet as the backbone in the related work part.
>
> Q4: There are many more papers on 10% ImageNet with much better results than DeiT. Even ignoring unlabeled data (as used by SimCLR, MoCo, ...), the S4L paper shows a ResNet50 trained on 10% gets 56%. The 50% and 80% ImageNet are almost pointless settings, but 1% is also widespread and missing.
>
>
> A4: Thanks for your kind suggestion. We add 1% and 5% imagenet experiments.
>
> Q5: What does "the landscape of the input" mean?
>
> A6: We meant to say: “And finally the MLP-Mixer also uses a static affinity matrix which changes based on the location of the pixels within the image.”. We will reword this to make it clearer.
>
> Q7: It's a stretch to call them akin to convolutions since they don't actually apply across locations.
>
> A7: Thanks for pointing out this. We will revise Line 303.
>
> [1] High-performance large-scale image recognition without normalization, Brock, Andrew, arXiv 2021

---

> > ### Comment · Reviewer_wcp4 · 2021-08-29
> > **Still good**
> >
> > Thanks for your answers. I still think this is good work and should be accepted, even after reading the other reviews and answers.

---

### Official Review · Reviewer_sSVS · 2021-07-17

**Rating:** 6
**Confidence:** 3

**Summary:**

This paper first provides a unified view on three existing network architectures: depthwise  CNNs, Transformers and MLP-Mixers. It then proposes Container and a light-weight version Container-Light, which combine CNNs and self-attention to aggregate spatial information. Experiments on ImageNet classification and COCO object detection verify the high modeling ability of Container.

**Limitations And Societal Impact:**

1. Container model adopts global self-attention modules. For image classification task, it requires much more computation (higher FLOPs) than other models with comparable parameters. For detection task, it is hard to directly applied and have to use the light-weight version.

2. The proposed model simply combines self-attention and depthwise convolution in one block, which lacks novelty in a degree.  With the unified framework of context aggregation, more brand-new modules can be developed. However, this part is missed in this paper.

**Main Review:**

This paper first proposes a unified spatial context aggregation framework for vision models, and points out that depthwise convolutions, Transformers and MLP-Mixers are special cases of this framework. Then it introduces Container blocks which employ self-attention and convolutional modules at the same time for context aggregation.

The discussion about how to unify different architectures and create new extensions in Section 3.2 and Section 4.1 are interesting. This discussion may help researchers to understand relationships between different architectures and inspire the community to create new ones

The proposed Container model achieves reasonably high classification accuracy and it consistently boost object detection performance with different frameworks, by a large margin.


**Time Spent Reviewing:**

6

---

> ### Author Response · Authors · 2021-08-09
> **Response to Reviewer sSVS**
>
> Q1: For image classification tasks, it requires much more computation (higher FLOPs) than other models with comparable parameters. For the detection tasks, it is hard to directly apply and have to use the lightweight version.
>
> A1: Container needs high computation due to the global self-attention. However, we performed the following ablation study and noticed that a good performance is achieved even without the global self-attention on high-resolution feature maps.
>
> | Methods | TOP-1 | Params | FLOPs | Throughput |
> | :-: | :-: | :-: | :-: | :-: |
> | Stage-1-2-3-4 (Container)   | 82.7 | 22.1 | 8.1 | 347.8 |
> | Stage-2-3-4 | 82.7 | 21.8 | 4.4 | 606.6 |
> | Stage-3-4  | 82.8 | 21.6 | 3.6 | 864.2 |
> | Stage-4  (Container-Light)   | 82.0 | 20.0 | 3.2| 1156.9 |
>
> From the above table, we find that we can reduce the computation of Container significantly by switching off the global self-attention in high-resolution feature maps without hindering the performance. In table 1, we compare SWIN-T and PVT-S. The results show that Container-Light can achieve better performance with fewer FLOPs and parameters.
>
> Besides, in table 4 and table 5, we show that even a lightweight version of Container can achieve superior performance than a strong baseline like SWIN-T, PVT-S, and ViL-S on Mask-RCNN and Retinanet.
>
> Q2: The proposed model simply combines self-attention and depthwise convolution in one block, which lacks novelty to a degree. With the unified framework of context aggregation, more brand-new modules can be developed. However, this part is missed in this paper.
>
> A2: The main novelty of this paper is to show the deep connection between different popular architectures and unify them using Context Aggregation. Built on this simple view, we propose a very simple and clean network architecture, Container,  which can achieve strong performance compared with strong baselines. We consider Container as a new starting point for future architecture development thus we keep the model as simple as possible. We will explore novel architectures motivated by the unified view of Container in the future.

---

### Official Review · Reviewer_ecxB · 2021-08-02

**Rating:** 5
**Confidence:** 5

**Summary:**

The paper proposes an architecture that combines Self-Attention, Mixer and DWise Conv blocks for spatial aggregation/mixing; along with channel mixing MLPs. While there's a lot of mathematical notation and it was a bit hard for me to understand quickly, my understanding is that: basically, the paper says - you can combine self-attention (dynamic filters generated) with DWise Conv (static + local) OR self-attention with Mixer (static + global) - and calls the two models as Container and Container-PAM (PAM = Pay Attention to MLP). Paper also proposes a hybrid model called Container-Lite which is basically Container with first three stages just using DWise Conv and no self-attention (for speeds) - and they use this architecture for detection/segmentation (in a manner similar to BoTNet). There are plenty of experiments in the paper - ImageNet, COCO, Self-Supervised Learning, etc. and some unifying presentation around MLP-Mixer, ViT and DWise Convs. Main review in the review section.

**Main Review:**

Positives:
1. Good empirical results.
2. Many benchmarks evaluated.

Negatives:

0. Unnecessary mathematical and verbose presentation of the idea. The core idea is extremely simple - just combine Mixer & Attention (or DWise and Attention). A simple diagram would go a long way. But please do correct me if my understanding is incorrect.
1. Citation to gMLP - which is also a hybrid model with mixer + attention makes sense to add.
2. Mixer - does share params across channels - "no parameter sharing" doesn't typecheck.
3. Improvements are compared wrt DeiT, a single-scale model, but since Container is doing multi-scale modeling, with engineered # of blocks like [2, 3, 8, 3] (similar to Swin), shouldn't you ideally be comparing to Swin? Container seems to be worse than Swin-S, but Container-Lite is better than Swin-T -- however Swin-T and Container-Lite don't seem directly comparable either - given Swin-T is pure attention and Container-Lite is like BotNet (3 stages conv and 1 attention) - it's expected that hybrids could also improve Swin-T right?
4. Dwise vs GConv - why does the design use DWise - and also - whats the hardware for the throughput measurements?
5. if Container relies heavily on Dwise, how does it scale to larger models? It's known DWise doesn't scale well to large models (ex EfficientNets-B6,7, L2, etc).
6. How much can a pure Mixer trained with all the data augmentation achieve? (in order to know the contribution of the hybrid design) -- FLOPs/param matched.
7. What about pure Dwise - with same design? - FLOPs/param matched.
8. BoTNet throughputs are easy to report since implementation is just ResNet with final stage using SA blocks. Would recommend authors to do it for ImageNet table at least.
9. Why no throughput numbers for Mask R-CNN results? Would be interested in knowing them since EfficientNets (and Dwise convs) are really slow for larger images.

I think paper has more negatives right now, but I am open to hearing back from authors.

**Time Spent Reviewing:**

2

---

> ### Author Response · Authors · 2021-08-09
> **Response to Reviewer ecxB**
>
> Q1:Unnecessary mathematical and verbose presentation of the idea.
>
> A1:The idea is quite simple which we believe is one of the advantages of this work. However, we provided mathematical formulations to formally present the idea. We can move them to the supplementary if the reviewers believe they are not necessary.
>
> Q2:Citation to gMLP[1]
>
> A2:gMLP proposes to use Mixer as a gate to modulate the features to replace the Transformers. The idea has been tested on image classification and BERT. Besides, gMLP shows that combining static Mixer weight with dynamically generated attention weight can improve the performance of BERT. gMLP is a concurrent work and we will add the discussion of gMLP.
>
> Q3:Mixer - does share params across channels?
>
> A3:In line 197, “No parameter sharing” means no parameter sharing between spatial locations like convolution operators. We will revise that.
>
> Q4:Why not comparing with Swin[2] since DeiT is single-scale?
>
> A4:In Table 1, we compare many Transformer-based models with multi-scale modelings, like PVT and Swin. To make the comparison clear, we show part of the table below:
>
> | Model | PVT-S | SWIN-T[2] | Container-Light |
> | :-: | :-: | :-: | :-: |
> | Param(M)  | 24.5  | 29 | 20 |
> | FLOPs(G) | 3.8 | 4.5 | 3.2 |
> | Top-1    | 79.8 | 81.3 | 82.0 |
>
> From the above table, we can conclude that compared with multi-scale modelings like PVT and Swin, Container-Light can achieve better performance with fewer parameters and FLOPs.
>
> Q5: Can a hybrid model improve Swin-T[2]?
>
> A5: Thank you for the suggestion. We are still working on this newly proposed experiment.
>
> Q6: Dwise vs GConv - why does the design use DWise - and also - what's the hardware for the throughput measurements?
>
> A6:In Line 173, we state that in Container only spatial information is propagated and no cross-channel information exchange happens. In contrast, GConv simultaneously aggregates spatial and partial channel information. The cross-channel exchange in GConv is redundant as the 1 by 1 convolution before and after GConv already performs cross-channel information fusion. The experiment in table 2, Group-Conv Vs DW-3 validates our hypothesis as GConv uses much more parameters and obtains lower performance. We use one V100 for throughput measurements.
>
> Q7:- if Container relies heavily on Dwise, how does it scale to larger models? It's known DWise doesn't scale well to large models (ex EfficientNets-B6,7, L2, etc).
>
> A7:We train a Container-Medium model with 57M parameters. We compare that with strong Transformer-based SWIN, convolution-based EfficientNet, and Mixture model CoAtNet (concurrent work).
>
> |          | TOP-1(%) | Params(M)  | FLOPs(G) | Throughput |
> | :-: | :-: | :-: | :-: | :-: |
> | Container-Light  | 82.0  | 20 | 3.2 |1156.9|
> | Container-M | 84.5 | 57 | 11.5 |340.5|
> | :-: | :-: | :-: | :-: | :-: |
> | Swin-T [2] | 81.3  | 29 | 8.7 |755.2|
> | Swin-B [2]  | 83.3 | 88 | 15.4 |278.1|
> | :-: | :-: | :-: | :-: | :-: |
> | Efficient-B4[4]  | 82.9  | 19 | 4.2 |349.4|
> | Efficient-B7[4]    | 84.3 | 66 | 37.0 |55.1|
> | :-: | :-: | :-: | :-: | :-: |
> | CoAtNet-0[3]  | 81.6  | 25 | 4.2 | N/A|
> | CoAtNet-3[3]    | 84.5 | 168 | 34.7 | N/A|
>
> From the above table, we can see that the scalability of Container is better than SWIN, EfficientNet, and CoAtNet. We will update the results and release a pre-trained Container-M model. Applying Container-M to SMCA-DETR, we observe significant improvements of mAP in the initial epochs. We will update the performance of the Container-M on SMCA-DETR later.
>
> Q8: How much can a pure Mixer trained with all the data augmentation achieve? (in order to know the contribution of the hybrid design) -- FLOPs/param matched.
>
> A8: Thank you for the suggestion. We are still working on this newly proposed experiment.
>
> Q9: What about pure Dwise - with the same design? - FLOPs/param matched.
>
> A9: In table 2, we report pure Dwise (row DW-3) which achieves 80.1% top-1 accuracy with 18.7M and 3.05G FLOPs. Container-Light achieves 82.0% top-1 accuracy with 20M and 3.2G FLOPs.
>
> Q10:  BoTNet throughputs are easy to report since the implementation is just ResNet with the final stage using SA blocks. Would recommend authors to do it for the ImageNet table at least.
>
> A10: The throughput of BoT-S1-50 measured on one V100 GPU using our own implementation is 874.65.  Note that S1 in Bot-S1-50 stands for stride 1 before the final stage.
>
> Q11: Why no throughput numbers for Mask R-CNN results? Would be interested in knowing them since EfficientNets (and Dwise convs) are really slow for larger images.
>
> A11: We compare Mask R-CNN with ResNet50 / Swin-T and Container-Light as the backbone.
>
> |          | ResNet50| Swin-T[2]   | Container-Light |
> | :-: | :-: | :-: | :-:|
> | FPS  | 17.3  | 14.6 | 14.1 |
> | Box mAP| 38.2 | 43.7 | 45.1 |
> | Mask mAP| 34.7 | 39.8 | 41.3 |
>
> [1] Pay Attention to MLPs, Liu, Hanxiao and Dai, Zihang and So, David R and Le, Quoc V, arXiv 2021
>
> [2] Swin Transformer: Hierarchical Vision Transformer using Shifted Windows, Ze Liu, arXiv 2021
>
> [3] CoAtNet: Marrying Convolution and Attention for All Data Sizes, Dai, Zihang, arXiv 2021
>
> [4] Efficientnet: Rethinking model scaling for convolutional neural networks, Tan, Mingxing, ICML 2019

---

### Decision · Program_Chairs · 2021-09-27

**Decision:**

Accept (Poster)

**Comment:**

The paper is somewhat borderline: three reviewers argue for acceptance, while one leans towards rejection. Based on the reviews, the rebuttal, the discussion, and the paper itself, below is the summary of key pros and cons.

Pros:
1) An interesting framework for unifying several classes of vision models (ConvNet, ViT, Mixer).
2) Good empirical results in terms of performance/compute tradeoff on classification and, especially, on detection.

Cons:
1) Not overwhelmingly novel
(some other points authors successfully addressed in their responses and I strongly recommend to include these responses/results in the paper)

To conclude, this is a solid paper, and the authors quite successfully responded to the concerns of the only negative reviewer, so overall I recommend acceptance.